# Posterior Reversible Encephalopathy Syndrome after Lenvatinib Therapy in a Patient with Olfactory Neuroblastoma

**DOI:** 10.3390/brainsci13010033

**Published:** 2022-12-23

**Authors:** Yu-Ju Tseng, Chun-Nan Chen, Ruey-Long Hong, Woon-Man Kung, Abel Po-Hao Huang

**Affiliations:** 1Department of Pharmacy, National Taiwan University Hospital, Taipei City 100, Taiwan; 2Department of Otolaryngology, National Taiwan University Hospital, Taipei City 100, Taiwan; 3Department of Oncology, National Taiwan University Hospital, Taipei City 100, Taiwan; 4Department of Exercise and Health Promotion, College of Kinesiology and Health, Chinese Culture University, Taipei City 111, Taiwan; 5Division of Neurosurgery, Department of Surgery, Taipei Tzu Chi Hospital, Buddhist Tzu Chi Medical Foundation, New Taipei City 231, Taiwan; 6Department of Surgery, National Taiwan University Hospital, Taipei City 100, Taiwan

**Keywords:** posterior reversible encephalopathy syndrome, lenvatinib, olfactory neuroblastoma

## Abstract

Posterior reversible encephalopathy syndrome (PRES) is a rare but severe neurological syndrome that may stem from the use of some medications. Although its mechanism is not well-known, hypertension and endothelial dysfunction have been mentioned in previous literature as being related. Lenvatinib serves as a neoplastic agent that inhibits the tyrosine kinase of vascular endothelial growth factor receptors (VEGFR). VEGFR inhibitors result in endothelial dysfunction and consequent hypertension by nitric oxide pathway suppression and endothelin (ET)-1 stimulation. We hypothesized that VEGFR inhibitors would cause PRES. Herein, we report the case of a 40-year-old man with olfactory neuroblastoma who developed PRES while undergoing treatment with lenvatinib, 7 months after initiation. The symptoms included loss of consciousness and seizures. Fortunately, the symptoms and presence of PRES in imaging resolved, 7 days and 1 month, respectively, after cessation of lenvatinib.

## 1. Introduction

Posterior reversible encephalopathy syndrome (PRES) is a severe but rare neurologic disorder characterized by headache, visual loss, seizure, elevated blood pressure (BP), and encephalopathy, which could be life-threatening [1]. The prevalence was reported in 0.03% of 21 million people in nationwide inpatient sample data from 2016 to 2018, indicating its rarity [2]. The typical clinical features related to the syndrome are cerebral edema and elevated BP, which can be evaluated by magnetic resonance imaging (MRI) findings on T2-weighted and fluid-attenuated inversion recovery (FLAIR) images [3] and by measuring BP. Some portions of these features are reported to be associated with the use of immunosuppressants and antineoplastic agents [4].

Antineoplastic tyrosine kinase inhibitors (TKI), including the vascular endothelial growth factor receptors (VEGFR) inhibitor lenvatinib, have been approved for the treatment of thyroid cancer, renal cell carcinoma, and hepatocellular carcinoma [5]; it has also been used for salvage therapy [6]. Lenvatinib may induce hypertension and PRES [4,7]—the mechanisms for the onset of PRES are still unclear but might be associated with pronounced physiological hypertension and endothelial dysfunction [8]. The VEGF signaling pathway induces nitric oxide (NO) production [9] and leads to the dilation of the blood vessels. The VEGFR inhibitors block the NO release and vasodilation mechanism, generating vasocontraction and, thus, causing the rise of BP. An acute increase in BP can disturb the cerebral blood flow regulatory mechanism, resulting in vasogenic edema [10], causing the typical features of PRES. Therefore, the management of blood pressure is considered an important controlling factor for the syndrome.

Despite PRES cases induced by lenvatinib used for thyroid cancer having been noted [11], lenvatinib therapy in patients with off-label cancer types is less reported. In the present study, a 40-year-old man with olfactory neuroblastoma treated by lenvatinib developed PRES, but the symptoms resolved under the discontinuation of lenvatinib and proper BP control.

## 2. Case Description

A 40-year-old man was diagnosed with olfactory neuroblastoma in 2012. The patient has since undergone multiple operations, radiotherapy, and chemotherapy. Lenvatinib 10 mg QD was prescribed for salvage therapy on 11 August 2020, after the cancer did not respond to other treatments. His relevant medical history of hypertension was controlled with amlodipine, along with multiple antihypertensive agents (Figure 1). On 9 January, 2021, a sudden change in consciousness, caused by seizures, was noted. Computed tomography indicated brain metastasis and brain abscess. Lenvatinib was suspended to prevent adverse reactions, and the patient was then treated with levetiracetam for seizure control and with aggressive antibiotics to cope with the brain abscess. After discontinuing lenvatinib, his BP was well controlled (120–140/80–90 mm Hg) with amlodipine and doxazosin.

On 26 January 2021, the patient underwent abscess drainage and progressed tumors were observed. Lenvatinib 10 mg QD was reinitiated for the progressed tumors, but the patient’s BP elevated to 150–170/100–120 mm Hg on 28 January, after the prescription. On 12 February, the involuntary movement was noted again. With the persistence of the abscess, broad-spectrum carbapenem was considered, despite the known side effects of seizures. Meropenem was administered from 9 to 13 February and from 15 to 22 February. On 18 February, brain MRI revealed a residual abscess in the right inferior frontal lobe, and hydrocephalus, ventriculitis, and meningitis were diagnosed. Since the abscess issue did not resolve, doripenem was administered from 22 February to 9 April. Electrolyte imbalances were corrected and metabolic disorders, other than hypertension, were treated.

The frequency and intensity of seizures gradually increased between 28 February and 4 March. On 2 March, his Glasgow Coma Scale score fell from E4V1M5 to E2V1M4. Lacosamide was administered for seizure control. Despite the use of aggressive antihypertensive agents, the patient’s BP remained poorly controlled (150–210/100–130 mm Hg). On 5 March, MRI indicated new bilateral occipital and bilateral high parietal cortico-subcortical lesions with low T1-weighted signal intensity and high T2-weighted signal intensity (Figure 2).

Given the clinical symptoms of hypertension and seizure following the MRI revelation of the bilateral occipital cortico-subcortical lesion, Grade 3 PRES was suspected [12]. Therefore, lenvatinib was discontinued on 8 March. However, a combination of afatinib 40 mg QD and lenvatinib 10 mg QD was subsequently indicated for tumor progression that was observed on 11 March. With the continued use of antitumor drugs, seizures occurred again on 12 March. Because of the poor wound healing after surgery, afatinib and lenvatinib were suspended on 22 March.

To cope with PRES, BP was controlled using amlodipine and labetalol. An MRI on 7 April indicated that PRES had resolved (Figure 3), with the clearance of brain edema. The patient resumed the use of lenvatinib on 18 August 2021, after 4 months of observation, with adequate blood pressure control. No subsequent episode of PRES was detected following regular follow-up MRIs. We also used the Naranjo algorithm to evaluate the causal relationship between the adverse event and the drug, which indicated that the use of lenvatinib was possibly associated with the syndrome. 

## 3. Discussion

PRES is a reversible neurological syndrome, presenting with headaches, nausea, dizziness, altered alertness, seizures, and visual disturbances. It characteristically manifests in neuroimaging as posterior cerebral white matter edema [5]. The disease grade of normal findings is scored as 0; faint or small abnormal areas are scored as 1; easily perceptible abnormalities are scored as 2; and large confluent areas of high-signal-intensity abnormality on T2-weighted and FLAIR images are scored as 3 [3]. In this case report, the large and high signal abnormality of the MRI images contributed to the diagnosis of Grade 3 PRES. 

The pathogenesis of PRES is not well known Possible proposed mechanisms are autoregulatory failure caused by hypertension, cerebral ischemia, and endothelial dysfunction, as shown in Figure 4. Other metabolic disturbances that may cause vasogenic edema have also been implicated [13,14,15]. Unlike cerebral ischemia or hemorrhage stroke, cerebral edema of PRES is reversible. VEGF signaling promotes endothelial-derived NO production. VEGFR inhibitors thereby cause vasoconstriction, increased vascular resistance, and hypertension. Alternatively, VEGFR inhibitors lead to endothelial dysfunction, increasing the release of endothelin-1 (ET-1), a potent vasoconstrictor that can cause hypertension [7,16]. In addition, inhibition of the VEGF pathway causes capillary rarefaction, which is a minor contributor to hypertension. Eight FDA-approved VEGFR inhibitors have warnings attached about the risk of PRES, including risk associated with lenvatinib. A study on lenvatinib observed hypertension in 45–73% of participants [17]. A hypothesis is that endothelial dysfunction and consequent hypertension lead to PRES. According to previous literature, anti-VEGFR TKI-induced PRES was highly associated with hypertension (Table 1). This implies that we should control blood pressure strictly for those patients under anti-VEGFR TKI. The National Cancer Institute recommends that BP be reduced to less than 140/90 mm Hg in treatment with VEGFR inhibitors (130/80 mm Hg in chronic kidney disease and diabetic patients) [18]. 

There is no definitive diagnostic criterion for PRES. Usually, it is diagnosed by exclusion of symptoms or lab data. If the patient has a new neurologic symptom (headache, visual symptoms, confusion, or seizure) and is concurrent with a specific clinical setting (hypertension, immunosuppressive agents, cytotoxic agents, or kidney disease), a brain MRI might be arranged. MRI, especially in diffusion-weighted imaging (DWI) sequence, could support the diagnosis [3]. Other than medication-induced hypertension, hypertensive encephalopathy and eclampsia could also lead to PRES. In addition, reversible cerebral vasoconstriction syndrome (RCVS) also owned many but not all pathogenic and clinical manifestations with PRES [19]; however, there was no evidence of arterial constriction documented by imaging in our patients.

Management of PRES is the same as that of essential hypertension. The use of angiotensin system inhibitors may be preferential, due to their beneficial effects on plasminogen activator inhibitor-1 expression and proteinuria, increasing the release of endothelial NO. According to the manufacturer, PRES occurred in 0.3% of the 1823 patients who received lenvatinib as a monotherapy [17]. The majority of cases of VEGFR inhibitor-induced PRES were reportedly associated with sorafenib [4]. Osawa et al. reported the first case of lenvatinib-caused PRES associated with uncontrolled BP [11]. The 66-year-old woman reported hypertension after 19 days and visual defects after 29 days of the initiation of lenvatinib. She was diagnosed with PRES by MRI and discontinued lenvatinib. Symptom relief and the appearance of PRES in imaging resolved after 1 month and 7 days of interruption. 

In our study, the patient’s BP increased from baseline approximately 1 month after lenvatinib was initiated. Although the first seizure episode occurred after 4 months, he was diagnosed with PRES by MRI 7 months after initiation. The etiology of his seizures was difficult to differentiate because of his operation history and metabolic disorders. The appearance of PRES in imaging resolved 1 month after the discontinuation of lenvatinib, which was similar to findings reported in previous case reports [11,20].

## 4. Conclusions

With the increasing use of lenvatinib in treating various malignancies, due to its specific pharmacodynamic properties both on-label and off-label, the incidence of adverse drug reaction is enhanced. One fatal side effect of PRES can be controlled and reversed under proper neurointensive care. The patient’s BP must be monitored and controlled to prevent the development of PRES with the prescription of lenvatinib.

## Figures and Tables

**Figure 1 brainsci-13-00033-f001:**
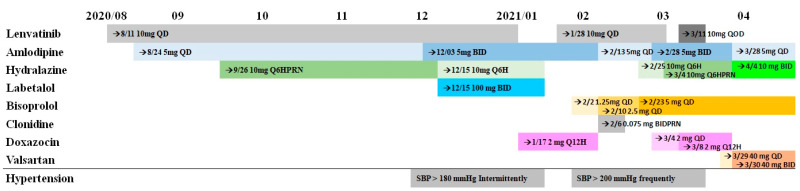
Timeline of the lenvatininb and anti-hypertensive agents use.

**Figure 2 brainsci-13-00033-f002:**
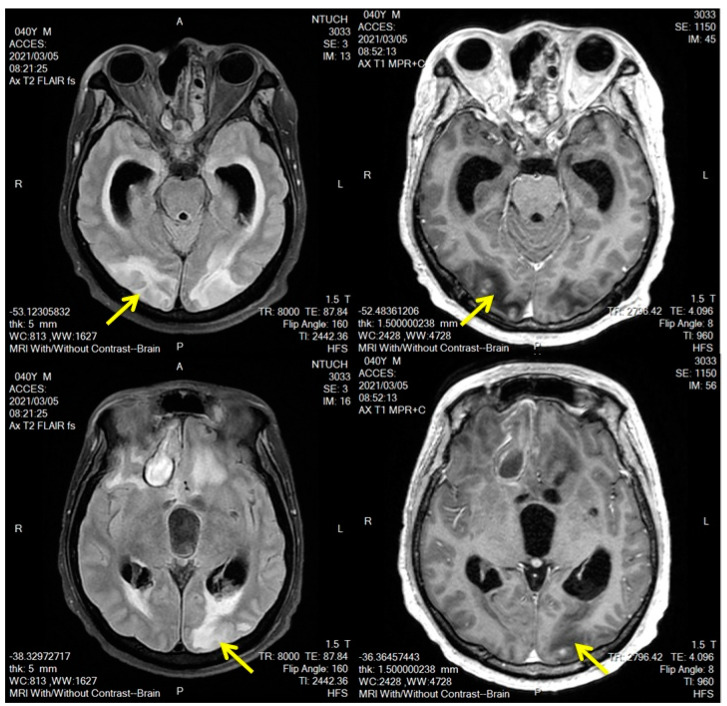
MRI was conducted on 5 March 2021. The yellow arrow indicates a bilateral occipital cortico-subcortical lesion with low T1-weighted SI, high T2-weighted SI, and faint scattered foci of enhancement.

**Figure 3 brainsci-13-00033-f003:**
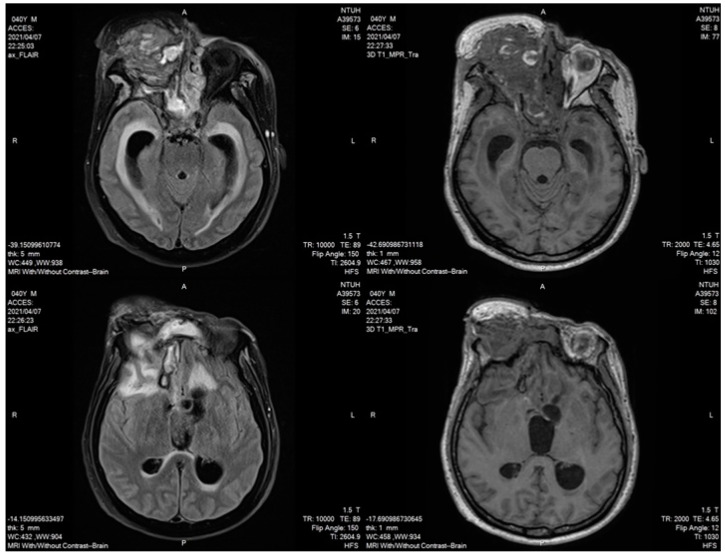
MRI was conducted on 10 April 2021. Subcortical lesions were resolved at bilateral occipital lobes.

**Figure 4 brainsci-13-00033-f004:**
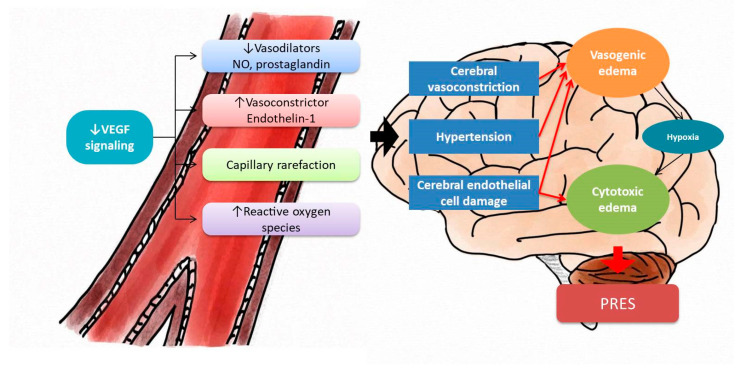
The hypothesis of anti-VEGF signaling induces PRES. Anti-VEGF signaling causes increasing peripheral resistance and, consequently, leads to hypertension. In addition, the anti-VEGF pathway results in endothelial dysfunction. Both hypertension and endothelial dysfunction damage the blood–brain barrier and increase vasogenic edema, and further cause cytotoxic edema by hypoxia. Endothelial cell damage itself contributes to cytotoxic edema. The sum of vasogenic edema and cytotoxic edema leads to PRES. VEGF = vascular endothelial growth factors; PRES = posterior reversible encephalopathy syndrome.

**Table 1 brainsci-13-00033-t001:** Case reports of TKI-associated PRES from literature after 2000.

	No. of Patients Reported	Targets	Hypertension before Medication (%)	Range of Onset Times(Months)	Symptoms	Hypertension after Medication (%)	Range of Recovery after Discontinuing (Symptoms, Days/Image, Months)	References
Anlotinib	1	VEGFR, Kit, PDGFR-α, FGFR.	0%	6	Headache, vomiting, mental confusion, and slurred speech.	100%	5/3	[21]
Apatinib	2	VEGFR2, Kit, and c-SRC.	0%	0.42–3	Dizziness, hseadache, blurred vision, diplopia, nausea and vomiting, lower limb weakness, and seizure.	100%	7/1–6	[22,23]
Axitinib	2	VEGFR, KIT PDGFR, and BCR-ABL.	50%	0.5–0.6	Headache, nausea, seizure, and loss of consciousness.	100%	7/0.25	[24,25]
Cabozantinib	1	AXL, FLT-3, KIT, MER, MET, RET, ROS1, TIE-2, TRKB, TYRO3, and VEGFR.	100%	0.75	Nausea, vomiting, confusion, and seizures.	100%	2/--	[26]
Cediranib	3	VEGFR1, VEGFR2, VEGFR3, and Flt-1.	33%^*^	2.5 *	Abdominal pain, obstipation, vomiting acute respiratory decompensation, and confusion. *	33% *	--/0.5 *	[27,28,29]
Cetuximab	1	EGFR, HER1, c-ErbB-1.	0%	2 h later	Headache, altered mental status, and generalized tonic-clonic seizures.	100%	3/2	[30]
Pazopanib	8	VEGFR, PDGFR-α, PDGFR-β, FGFR-1, FGFR-3, KIT, Lck, and c-Fms.	29%	0.1–2	Headache, nausea, vomiting, impaired vision, gait instability, pitting edema, seizure, impaired consciousness, and s status epilepticus.	100%	2–7/0.5–3	[31,32,33,34,35,36,37,38]
Regorafenib	3	VEGF, KIT, PDGFR, RET, FGFR, TIE2, DDR2, TrkA, Eph2A, RAF-1. BRAF, BRAF^V600E^, SAPK2, PTK5, and Abl.	33%	0.1–3	Agitation, headache, vomiting, mental status changes, reduced hand coordination and dexterity, a homonymous inferior quadrantanopia, and seizures.	67%	7/1 *	[39,40,41]
Ruxolitinib	1	JAK 1 and JAK 2.	0%	12	Altered mental status, focal seizure, bowel incontinence, and unsteady gait.	100%	2/--	[42]
Sorafenib	6	CRAF, BRAF, VEGFR, PDGFR-β, KIT, FLT-3, RET, and RET.	50%	0.25–4	Headache, vertigo, loss of vision, hallucinations, hypersalivation, seizure, and loss of consciousness.	83%	1–33/0.25–1.2	[43,44,45,46,47]
Sunitinib	11	PDGFR, VEGFR, FLT-3, CSF-1R, and RET.	36%	0.25-over 4 years	Headache, vision loss, gait unsteadiness, dizziness, weakness, optic ataxia, abdominal pain, seizure, and lack of consciousness.	100%	2–21/0.25–2	[48,49,50,51,52,53,54,55,56,57,58]
Vemurafenib	3	BRAF/MEK.	0%	0.4–2	Confusion, headache, nausea, vomiting, altered mental status, impaired speech, and seizure.	67%	7/0.36–2.5	[59,60,61]
Lenvatinib	7 (including our case)	VEGFR, FGFR, FGFR 4, PDGFR-α, KIT, and RET.	75%	2 h–7	Headache, visual field defect, forgetfulness, incoherent speech, restlessness mimicking dementia, and seizures	100%	1–14/0.25–1	[11,17,20,62]

No. = number; VEGF = vascular endothelial growth factor; PDGFR = platelet-derived growth factor receptor; FGFR = fibroblast growth factor receptors; KDR = kinase insert domain receptor; c-SRC = proto-oncogene tyrosine-protein kinase Src; FLT-3 = Fms-like tyrosine kinase-3; IL = interleukin; Lck = lymphocyte-specific protein tyrosine kinase; c-Fms = transmembrane glycoprotein receptor tyrosine kinase; JAK = Janus kinase; CSF-1R = colony-stimulating factor type 1; RET = glial cell-line-derived neurotrophic factor receptor. * Only one case mentioned the detail.

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
