# Peer review of "Posterior Reversible Encephalopathy Syndrome after Lenvatinib Therapy in a Patient with Olfactory Neuroblastoma"

_brainsci, 2022, doi:10.3390/brainsci13010033_

Round 1
Reviewer 1 Report
I have read the manuscript (Posterior Reversible Encephalopathy Syndrome After Lenvatinib Therapy in a Patient with Olfactory Neuroblastoma) very carefully. The authors present a neurooncological complicated with PRES after Lenvatinib Therapy. It is a well written manuscript with helpful figures. The topic is relevant, interesting, and original. The main question is to present a case with a rare complication of a Therapy with Lenvatinib in a patient with an Olfactory Neuroblastoma. Compared with other published material, it presents the current state of the case + the not well known complication.
I still have some comments though.
It's discussion may need argumentation if the hypertension per se could be the cause of PRES and not the medication itself, even if it is already mentioned in the drug information. And it still need more differential diagnosis from a neurological point of view.
Otherwise I wonder why there is no Neurologist or Neurosurgeon involved in the case presentation.
Author Response
Dear Reviewer, thank you for your valuable suggestions, please see the attachment for our responds.

Reviewer 2 Report
1. Please, improve the abstract description. Remember that in most indexations, only the abstract is provided. E.g., provide more specific information.
2. Please provide a table with all the cases already reported with PRES and lenvatinib.
3. Could the authors explain why they believed that PRES was associated with lenvatinib if the same drug was used for about five months before the presentation?
It is difficult to associate a causative drug when the patient uses many other drugs.
It is advised to calculate the Naranjo algorithm, including the result in the report.
4. It is advised to provide a figure about the proposed mechanism.
Author Response

(The authors gave the same response as above.)

Round 2
Reviewer 1 Report
Thank you for the response. I dont have further commments.
Reviewer 2 Report
Satisfactory